# Cytokinin-Controlled Gradient Distribution of Auxin in *Arabidopsis* Root Tip

**DOI:** 10.3390/ijms22083874

**Published:** 2021-04-08

**Authors:** Lei Wu, Jun-Li Wang, Xiao-Feng Li, Guang-Qin Guo

**Affiliations:** Institute of Cell Biology and MOE Key Laboratory of Cell Activities and Stress Adaptations, School of Life Sciences, Lanzhou University, Lanzhou 730000, China; leiwu@lzu.edu.cn (L.W.); wangjl19@lzu.edu.cn (J.-L.W.); xfli@lzu.edu.cn (X.-F.L.)

**Keywords:** cytokinin, auxin, biosynthesis/metabolism, auxin transport, cell division, cell differentiation, root meristem

## Abstract

The plant root is a dynamic system, which is able to respond promptly to external environmental stimuli by constantly adjusting its growth and development. A key component regulating this growth and development is the finely tuned cross-talk between the auxin and cytokinin phytohormones. The gradient distribution of auxin is not only important for the growth and development of roots, but also for root growth in various response. Recent studies have shed light on the molecular mechanisms of cytokinin-mediated regulation of local auxin biosynthesis/metabolism and redistribution in establishing active auxin gradients, resulting in cell division and differentiation in primary root tips. In this review, we focus our attention on the molecular mechanisms underlying the cytokinin-controlled auxin gradient in root tips.

## 1. Introduction

Roots are plant organs, usually located below the surface of the soil, where they grow and respond to various environmental stimuli. Plants receive essential water and nutrients from the soil through roots. Thus, overall plant survival depends on root growth and development. Auxin is the most important phytohormone that regulates the growth and development of plant roots [1,2,3,4,5]. Although auxin plays a central role in the control of root growth and development, much evidences have now been collected showing that the correct growth and development of roots also depends on auxin cross-talk with other phytohormones, such as cytokinin [6,7]. Since the discovery of cytokinin as an inducer of cell division in cultured plants, the function of cytokinin has been linked to that of auxin [8,9]. Earlier studies have suggested an antagonistic relationship between these two phytohormones; however, the truth is more complicated, with both antagonistic and supportive interactions that are usually cell and/or tissue-specific [10,11,12].

It is usually considered that the key to the function of auxin in plant growth and development is its gradient distribution [13,14]. Of all the mechanisms that regulate auxin distribution, cytokinin plays a prominent role, not only by regulating local auxin metabolism [15,16,17,18,19,20,21,22], but also by modulating PAT (polar auxin transport) [11,23,24,25,26,27]. In the last decade, multiple points of cross-talk between auxin and cytokinin, including biosynthesis/metabolism, transport, and signaling, have been revealed [12,23,28,29,30,31,32,33,34,35,36,37,38,39,40,41,42,43,44,45,46,47,48]. In this review, we focus on cytokinin-controlled gradient distribution of auxin by regulating its biosynthesis and transport, and its role in regulating root growth and development.

## 2. Cytokinin Signal Pathway

The cytokinin signaling pathway in plants is similar to the bacterial multi-step two-component signal transduction system [12,34,36,49]. In *Arabidopsis*, cytokinin binding leads to autophosphorylation of membrane-bound cytokinin receptors AHK2 (Arabidopsis histidine kinase 2), AHK3 and AHK4/CRE1 (cytokinin response 1), followed by a phosphorylation cascade [50,51,52,53].The phosphoryl group is transfered from receptors to AHPs (Arabidopsis histidine phosphotransferase proteins) [54,55,56,57], which enters the nucleus and phosphorylates the ARRs (Arabidopsis response regulators). ARRs can be divided into two types according to their structure. Phosphorylated type-B ARRs work as TFs (transcription factors), activating cytokinin-responsive genes [58,59,60,61]. Unlike the type-B ARRs, the type-A ARRs lack a DNA-binding domain, and their expression is rapidly induced by cytokinin, which forms a feedback loop by negatively regulating type-B ARRs [62,63,64,65]. Furthermore, some CRFs (cytokinin responsive factors), identified as AP2 TFs [66,67], also play a role in cytokinin-regulated gene expression [67].

## 3. Cytokinin-Regulated IAA Biosynthesis

Based on biochemical and genetic evidences, the major natural auxin in plants, IAA (indole-3-acetic acid), is synthesized via two major pathways: Trp (Tryptophan)-independent (TI) and Trp-dependent (TD) pathways [5,22,68]. So far, the molecular components of the TI pathway have been poorly understood [69].

At present, it appears that the best understood IPA (indole pyruvic acid) pathway is the main TD pathway of auxin biosynthesis in *Arabidopsis thaliana* [48,70,71], in which TAA (tryptophan aminotransferase of Arabidopsis) family proteins catalyze the conversion of Trp to IPA [16,72,73,74], and YUC (YUCCA) flavin monooxygenase-like proteins catalyze the conversion of IPA to IAA [70,71,75]. Overexpression of *YUCs*, but not *TAA* family genes, leads to auxin overproduction, implying that the YUCs, rather than TAA family proteins, catalyze the rate-limiting step of the IPA pathway [70,76,77,78].

Besides TAA1/WEI8/SAV3/TIR2/CKRC1 (weak ethylene insensitive 8/shade avoidance 3/transport inhibitor response 2/cytokinin induced root curling 1) [16,72,73,74], the TAA family also includes two other homologous proteins: TAR1 (tryptophan aminotransferase related 1) and TAR2, which have overlapping functions [72]. YUCs belongs to a large gene family with 11 members in the *Arabidopsis* genome, which are functionally redundant to each other [75,76,77,78,79,80,81]. The importance of the IPA pathway in plant growth and development has made it the focus of research in recent years, and it is regulated by various developmental and environmental signals, including phytohormones [22,48,82].

Early definite evidence of the effect of cytokinin on auxin biosynthesis was reported by Jones et al. (2010), showing that cytokinin can induce auxin biosynthesis and that some signaling components of auxin and cytokinin are involved in this process; however, the molecular mechanism remains unclear [15]. Later, the authors found that cytokinin can induce the expression of both *TAA1* and *YUC8* genes to enhance auxin production [16,17]. In the adventitious root apex of *Arabidopsis,* cytokinin-mediated up-regulation of *YUC6* was found to be involved in the formation of the QC (Quiescent Center) [83]. Cytokinin-induced expression of *YUC1* and *YUC4* in the gynoecia primordium has been reported to ensure correct domain patterning [39].

Transcriptional activation of *TAA* and *YUC* family genes by cytokinin is dependent on cytokinin signaling transduction, as has been shown for *TAA1/CKRC1* [16] and *YUC8/CKRC2* [17] in *Arabidopsis* (Figure 1). Yan et al. (2017) found that type-B ARRs can directly bind two cis elements in the promoter and the second intron of *TAA1* to activate its transcription [18]. The AHKs-ARR1/12-mediated cytokinin signaling pathway is also reported to be necessary for cytokinin-induced up-regulation of *TAA1* and *YUC8,* and the PIF4 (phytochrome-interacting factors 4) is required for this upregulation. Transcription of *PIF4* itself is induced by cytokinin via the AHKs-ARR1/12 signaling pathway, indicating that PIF4 plays an essential role in mediating the regulatory effect of cytokinin on the transcriptions of *TAA1* and *YUC8* genes in the IPA pathway of auxin biosynthesis [17].

Upstream of the IPA pathway, the two subunits of rate-limiting anthranilate synthase, *ASA1* (anthranilate synthase alpha-subunit 1)*/WEI2/CKRC6* and *ASB1* (anthranilate synthase beta-subunit 1)/*WEI7* in the Trp biosynthesis pathway [72,84], are specifically expressed in the root tip [72,73,79,85]. It has been shown that the transcriptional levels of *ASA1**/WEI2/CKRC6* and *ASB1**/WEI7* can be induced by cytokinin [15,85,86], and ARR1 promotes auxin biosynthesis in the stem cell niche via the up-regulation of *ASB1/WEI7* in the root meristem (Figure 1) [85].

In another proposed but still somewhat obscure TD pathway named the IAOx (indole-3-acetaldoxime) pathway, the key enzymes of cytochrome P450 proteins CYP79B2 and CYP79B3 convert Trp to IAOx. Overexpression lines of these two genes showed auxin overproduction phenotypes and a high level of auxin; on the contrary, plants showed auxin-deficient phenotype and decreased endogenous auxin content when the enzyme function was lost [87,88]. Expressed in the root meristem [87], *CYP79B2* and *CYP79B3* are involved in root elongation [87,88]. Microarray and qRT-PCR results show that their transcription is activated when the plant is treated with cytokinin [15,89], and the result of ChIP-seq (chromatin immunoprecipitation sequencing) shows that ARR1 can bind to a partial sequence of a *CYP79B3* gene after 3 h cytokinin treatment [90]. It is likely that the IAOx pathway could be another cytokinin-regulated auxin biosynthesis pathway.

Most recently, it was found that exogenous cytokinin can stimulate the expression of *CKRW2/HUB1* (cytokinin induced root waving 2/histone monoubiquitinate 1), which encodes an E3 ligase required for histone H2B mono-ubiquitination (H2Bub1) to promote the transcription of auxin biosynthetic genes *TRP2/TSB1* (tryptophan biosynthesis 2/tryptophan synthase beta-subunit 1)*, ASB1/WEI7, YUC7* and *AMI1* (*amidase 1*) [91]. This discovery reveals an epigenetic mechanism of cytokinin-regulated IAA biosynthesis at the chromatin level.

In conclusion, cytokinin can control the level of auxin in roots by regulating local auxin biosynthesis, which is generally believed to be the basis of the gradient distribution of auxin [48,72,74,92], and is necessary for root growth and development [82,93,94].

## 4. Cytokinin-Regulated IAA Conjugation and Degradation

In *Arabidopsis*, the GH3 (Gretchen Hagen 3) family belongs to a large gene family with three groups, of which group II (GH3.1-6, GH3.9 and GH3.17) has been shown to convert IAA to IAA amino acid conjugates [95,96,97]. The conjugation of amino acids with IAA is usually classified into two categories: one that can be converted to free IAA through hydrolysis and is considered to be related to the storage of auxin, such as IAA-Ala and IAA-Leu [98,99]; the other is believed to be related to the degradation of auxin, which can be irretrievably oxidized after formation and then degraded, such as IAA-Asp and IAA-Glu [100,101,102]. The process of IAA amino acid conjugation is generally considered to be associated with auxin homeostasis, which may play a role in cases where plant cells have to rapidly alter the relative amount of IAA in response to developmental and environmental changes [19,22,95,102,103,104,105,106].

The LRC (lateral root cap) is the outermost tissue of the root meristem [107]. If the LRC is lost, the size of the meristem will be greatly reduced [108,109]. This is because bPAT (basipetal PAT, from the root tip to the elongation zone) starts at the lateral root cap. If the LRC is lost, auxin transported from the root tip to the elongation zone will be disturbed. The defect of bPAT makes it impossible to establish a normal gradient distribution of auxin, so that the root meristem becomes smaller [21,110]. However, it is interesting to note that several members of group II GH3 (GH3.5, GH3.6 and GH3.17) are specifically expressed in the LRC [19,20,21], in which GH3.17 catalyzes IAA to IAA-Glu to participate in IAA degradation [19,21]. It has been reported that these three conjugation enzymes play a key role in controlling auxin flow in bPAT, as they determine the amount of auxin transport from the root tip to the elongation zone [19,20,21].

Surprisingly, *GH3.5, GH3.6* and *GH3.17* are downstream of type-B ARR1 in cytokinin signal transduction, and are targets of cytokinin–auxin antagonism [20,21]. Cytokinin suppresses bPAT by activating transcription of *GH3.5, GH3.6* and *GH3.17*, which convert free IAA to IAA amino acid conjugates, thus regulating the size of the root meristem (Figure 1) [20,21].

## 5. Cytokinin-Regulated Intercellular Auxin Transport

The carriers that mediate auxin transport between cells contain three protein families: (1) AUX1/LAX (AUX1/LIKE AUX1) family proteins, responsible for the transport of auxin from the apoplast into the cell [111,112,113,114,115]; (2) PIN (PIN-formed) family proteins that mediate auxin output cells [116,117,118,119,120]; (3) ABCB/PGP/MDR (ATP-binding cassette protein subfamily B/P-Glyco protein/multidrug resistance) family proteins, involved in the ATP-driven influx or efflux of auxin [121,122].

Of these three families, only AUX1/LAX influx and PIN efflux carriers are involved in PAT machinery, directing the flow of auxin from the shoot acropetally through the stele toward the root tip (aPAT, acropetal PAT). From here it is basipetally redistributed via the epidermis to the elongation zone (bPAT) [115,116,120,123,124,125,126,127]. The pattern of expression of the various AUX1/LAX and PIN genes and the localization of them on specific cell faces play a key role in PAT machinery to determine the distribution of auxin in plant tissues [115,116,120,123,124,125,126,127]. Unlike AUX1/LAX influx and PIN efflux carriers, the ABCB/PGP/MDR family proteins have also been shown to act as auxin transporters to mediate auxin in and out of cells; however, because they are uniformly localized in the cell, they are considered to be unrelated to PAT [128,129].

In the last 10 years, studies on cytokinin-regulated plant development have revealed that a number of processes are involved in cytokinin interaction with PAT (e.g., root and shoot apical meristem activity maintenance, lateral root organogenesis, vasculature differentiation, or phyllotaxis [11,26,47,130,131]). In primary roots, previous studies suggested that cytokinin inhibition of cell expansion depended on cytokinin-induced ethylene biosynthesis [132]. The inhibition of root cell elongation requires ethylene regulated transport-dependent auxin distribution [27,133]. Although the role of ethylene in the cytokinin response has been demonstrated, the direct regulation of PAT by cytokinin is more important for root growth and development.

### 5.1. PINs Efflux Carriers

In *Arabidopsis*
*thaliana*, according to the length of the hydrophilic loop in the middle of the polypeptide chain, the PINs family is divided into two subfamilies: as auxin efflux carriers, PIN1, PIN2, PIN3, PIN4 and PIN7 contain a long hydrophilic loop and are located in the PM (plasma membrane) [117,119,120], while PIN5, PIN6 and PIN8 with a short hydrophilic loop are mainly located in the ER (endoplasmic reticulum), which are involved in intracellular auxin transport [117,119,120,134]. All of the PIN efflux carriers are expressed and active in the root tip and perform their respective functions [116,117,119,120,129,135].

Cytokinin has been shown to influence cell-to-cell auxin transport by regulating the expression of several PIN genes, thereby modulating auxin distribution, which is essential for root development [11,26,131,136,137]. In *Arabidopsis* roots, through the cytokinin receptor AHK3 and the downstream signaling components ARR1 and ARR12, cytokinin has been shown to activate SHY2 (short hypocotyl 2), which is a member of the AUX/IAA (Auxin/Indole-3-Acetic Acid) protein family that heterodimerizes with ARFs (auxin response factors), preventing the activation of auxin responses. Therefore, as a downstream gene of AFRs, the expression of *PIN1*, *PIN3* and *PIN7* was inhibited when SHY2 was activated (Figure 1) [11,138].

Some CRFs also directly fine-tune PIN expression, providing a direct regulatory link between cytokinin signaling and the auxin transport machinery. Plants lacking CRF activity show developmental pattern aberrations consistent with abnormal auxin distribution. Removal of specific cis-regulatory elements (PCRE (PIN cytokinin response element) domain 5′-AGCAGAC-3′-like motif) effectively uncouples PIN1 and PIN7 transcription from the CRF-dependent regulation, and attenuates plant cytokinin sensitivity (Figure 1) [137]. Furthermore, the bHLH TF SPATULA enables cytokinin signaling, and activates the expression of *PIN3* [139].

Besides transcriptional regulation, cytokinin also negatively regulates PINs at the post-transcriptional level [140]. Cytokinin can affect endomembrane trafficking of PIN1, PIN3 and PIN7 to redirect them for lytic degradation in vacuoles to reduce their abundance on the plasma membrane [141], and this function relies on canonical cytokinin signaling components, including the cytokinin receptor AHK4/CRE1 and some type-B ARRs [47]. PIN1 phosphorylation status is also involved (Figure 1) [142].

The complexity of cytokinin effects on PINs to regulate PAT in various cells/tissues/organs and developmental stages by various mechanisms has led to some confusing or even seemingly contradictory results. For example, in studying the effect of exogenous cytokinin on the transcription of PIN1, cytokinin was found to inhibit PIN1 transcription by using a 2-mm root tip with meristem/transition/elongation zones as the material [140], but was reported to have no such effect on a 0.5-mm root tip mainly with meristem zones [11,26,140]. In fact, cytokinin still causes PIN1 inactivation in the meristem, which depends on cytokinin-induced post-transcriptional regulation [140]. As another example, cytokinin down-regulates PIN1 and PIN3 proteins in primary roots to inhibit aPAT [11,26,136], but promotes the accumulation of PIN3, PIN4, and PIN7 in shoots, thereby coordinating bud outgrowth and branching [47].

Despite these complications, at present, it is generally believed that cytokinin down-regulates PAT by inhibiting all PIN efflux carriers except PIN7 in primary root tips [11,26,93,136,137,140,141].

### 5.2. AUX1/LAX Influx Carriers

In *Arabidopsis*
*thaliana*, AUX1/LAX influx carriers are encoded by a small multigene family comprised of four members: *AUX1*, *LAX1*, *LAX2*, and *LAX3* [111,112,113,114,115]. They display reasonably distinct expression patterns and are suggested to participate in different developmental processes [111]. Of all the members, except for *LAX1*, which is not involved in root development [111], the *AUX1* gene is mainly expressed in the LRC, epidermal and phloem tissues near the root tip [113,143], and has been shown to play a role in gravitropism [143]; both *AUX1* and *LAX3* are shown to regulate lateral root development [112], and *LAX2* is strongly expressed in the QC and the LRC [111], where it plays a key role in maintaining the stem cell fate surrounding the QC [144]. It was found that disruption of the *LAX2* gene results in a phenotype similar to that observed in type-A ARR mutants, such as increased division of cells in the QC [144]. This is because auxin influx carriers, *LAX2* genes, act downstream of cytokinin in the root tip, whose transcription is suppressed by cytokinin [27,144]. The decrease in *AUX1* and *LAX2* expression in response to cytokinin requires cytokinin response transcriptional effector type-B ARRs, which mediate the primary transcriptional response to cytokinin (Figure 1) [27,144,145]. CHIP assays showed that the *AUX1* gene was enriched for extended type-B ARR12 binding motifs in intron 8 [27,145], and type-B ARR1 was found to bind directly to intron 2, intron 4 and 1.2 kb upstream motifs of the LAX2 gene [144]. These studies indicate that cytokinin response transcriptional effector type-B ARRs directly down-regulate the expression of AUX1/LAX influx carriers.

## 6. Cytokinin-Regulated Intracellular Auxin Transport

In addition to the above-mentioned PINs for intercellular PAT, the auxin carrier proteins for intracellular auxin transport include ER-localized PIN5, PIN6, PIN8, and other PILSs (PIN-like proteins), which are likely older than PINs by phylogenetic analysis [120,146,147,148]. There are seven known members of the PILS family. Although the PILS proteins share only 10–18% of their sequence with PIN proteins, they are topologically similar [147,148,149]. Members of the PILS family are identified by the presence of an auxin carrier domain that spans almost the entire length of the PILS proteins; therefore, PILS proteins still have the ability to transport auxin across the membrane [120,146].

Compared with the auxin efflux PINs located on the plasma membrane, which are involved in the intercellular transport of auxin, ER-localized PINs and PILSs mediate the intracellular transport of auxin [120,134,150,151,152,153,154]. ER-localized PINs are speculated to mediate auxin flow into (PIN5) or out (PIN8) of the ER lumen [120,152,154], or hypothetically from the ER lumen into the nucleus (PIN6 and PIN8) to open the auxin downstream genes’ transcription [150,152]. Like PIN5, the expression of PILS2 and PILS5 transporters causes cytosolic auxin to be transported into the ER lumen, leading to reduced transcriptional regulation of downstream genes by auxin in the nucleus, thus reducing auxin signals and cell sensitivity to auxin [120,146,147,148,155,156].

PIN5 is expressed in the vasculature of the mature root zone [157] and epidermis of the meristem zone [21]; PILS2 and PILS5 showed a particular overlapping expression in the root transition zone [146]. In root growth and development, PIN5, PILS2 and PILS5 play a negative role in primary root elongation [21,146,148]. The roots of PIN5, PILS2, or PILS5 gain-of-function mutants become shorter; on the contrary, the roots of loss-of-function mutants become longer [21,146,148]. This is because ER-localization auxin transport carriers, PIN5 and PILSs negatively regulate PAT and auxin signaling [21,134,146,147,154,155,156]. Interestingly, unlike PINs involved in intercellular transport, transcription of *PIN5* and *PILS5* is induced by cytokinin (Figure 1) [15,21,158,159], suggesting that PIN5 and PILS5 are other targets of cytokinin–auxin antagonism besides auxin conjugation enzymes GH3.5, GH3.6 and GH3.17. In other words, the process of cytokinin activating *PIN5* and *PILS5* expression reduces the amount of auxin transport to root elongation zones through intracellular auxin accumulation in bPAT transport cells. A recent paper reported that ARR1 binds directly to the *PIN5* promoter to mediate cytokinin induction of *PIN5* expression (Figure 1) [21]. Furthermore, it has been reported that PIN5-mediated intracellular auxin accumulation and GH3.17-mediated auxin conjugation with Glu are inextricably linked, which together regulate auxin homeostasis and signal transduction [21,160]; however, the details require further study.

## 7. Concluding Remarks

Once the seed germinates, the root meristem, which is derived from the proximal stem cells, proliferates and expands rapidly. After approximately 5 days of growth and development, the meristem of the root tip reaches a stable size, by which root growth is sustained. Cytokinin and auxin interactions play key roles in controlling the balance between the rate of cell division and differentiation, which is crucial for the maintenance of root meristems [11,19,26,161]. Here, we reviewed the cytokinin-mediated regulation on the components involved in auxin biosynthesis/metabolism, polar auxin transport, and intracellular transport of auxin in root tips (Figure 1). These processes are all related to establish the gradient distribution of auxin in root tips. They work together to control the size of the meristem, which is significantly reduced in *taa1/ckrc1* [16], *yuc8/ckrc2* [17], and the triple mutant *pin1 pin3 pin7* [11], but incteased in *pin5-3* [21], *gh3.5, gh3.6* and *gh3.17* [19,20,21].

According to the current auxin gradient distribution model, auxin concentration is maximal at the QC of the root tip, but minimal at the boundary of the transition zone (Figure 2) [19,162]. The auxin maximum is very important for the maintenance of the QC and stem cell fate surrounding the QC [162], while the auxin minimum at the boundary of the transition zone acts as a signal to control the developmental switch from cell division to cell differentiation [19]. Earlier studies believed that the establishment of the maximum value of auxin in the root tip depended on the auxin transport mediated by PINs. This auxin maximum involves three transport processes, which are aPAT, bPAT and reflux of auxin. The auxin reflux redistributes auxin to form a characteristic gradient, and auxin is maximally located at the QC [162]. IAA3/SHY2, a negative regulator of auxin signal transduction, is a direct transcriptional target downstream of cytokinin signal AHK3-ARR1,12, which is one of the sites where the two hormones interact [11,138]. Cytokinin signaling suppresses auxin efflux carriers PIN1, PIN3 and PIN7 expressions through IAA3/SHY2 upregulation [11,26], and at the same time, promotes PIN1, PIN3 and PIN4 protein degradation to decrease PIN abundance at the post-transcriptional level [140]. Thus, cytokinin controls the flow of auxin to the root tip through the precise regulation of PINs. This process not only determines auxin levels that generate a minimum in the vascular tissue of the transition zone, but also determines the maximum value of auxin in the QC of the root tip (Figure 2) [11,19,26,140,163].

In addition, the auxin reflux mechanism gradually decreases auxin when it is transported along the external tissues of roots from the top to the base [162]; however, the formation of the external tissue auxin gradient is also regulated by cytokinin-mediated auxin degradation [19]. Several years ago, fluorescence-activated cell sorting of green fluorescent protein (GFP)-marked cell types, combined with solid-phase microextraction and an ultra-high-sensitivity mass spectrometry (MS) assay, was applied to analyze the levels of cytokinin in each tissue of the root tip (Figure 2) [164]. Cytokinin was found to be mainly concentrated in the lateral root cap, columella, columella initials, QC cells, and in the epidermis of the root tip, forming a gradient down to a maxima at the root cap [164]. Several components involved in cytokinin regulating auxin gradient distribution are specifically expressed in high cytokinin level regions of the root. When the auxin in the root tip was transported from the LRC to the elongation zone through bPAT, cytokinin could up-regulate the expression of genes *GH3.5*, *GH3.6*, *GH3.17* and *PIN5* in the LRC and the epidermis near the root tip to gradually reduce the amount of auxin transported by bPAT (Figure 2) [19,21,160].

Although calculated models based on the PINs and other auxin transporters indicate that the transport and redistribution of auxin from the above-ground parts is sufficient to produce the auxin gradient and auxin maximum in the root [165], the effect of local auxin biosynthesis on growth and development cannot be ignored [81,82,94,163]. Disturbing local auxin biosynthesis in the root tip will cause developmental defects, such as the small meristem and gravitropic defect [16,17,86]. There are evidences that auxin overproduction in shoots cannot completely rescue root auxin deficiency phenotypes, highlighting the importance of local auxin biosynthesis for root development [79]. In Section 3, we reviewed several processes by which local auxin biosynthesis is activated by cytokinin. At present, little is known about the physiological functions of these processes. We speculate that the local auxin biosynthesis induced by cytokinin may be a compensation mechanism of auxin. The establishment and maintenance of the gradient distribution of auxin in the root tip depends on auxin transport and reflux, and auxin is constantly attenuated by the downstream signal of cytokinin in the flow (Section 4 and Section 6). This compensation mechanism can offset the attenuated auxin, thereby maintaining the stability of the meristem.

In summary, recent studies highlight the important role of the cytokinin-controlled gradient distribution of auxin in root growth and development. However, some questions remain. For example, there are many mechanisms by which cytokinins regulate the gradient distribution of auxin; how do plants balance the local auxin biosynthesis, degradation, intercellular transport and intracellular transport in this process? Are these processes connected or dissected during plant growth and development?

## Figures and Tables

**Figure 1 ijms-22-03874-f001:**
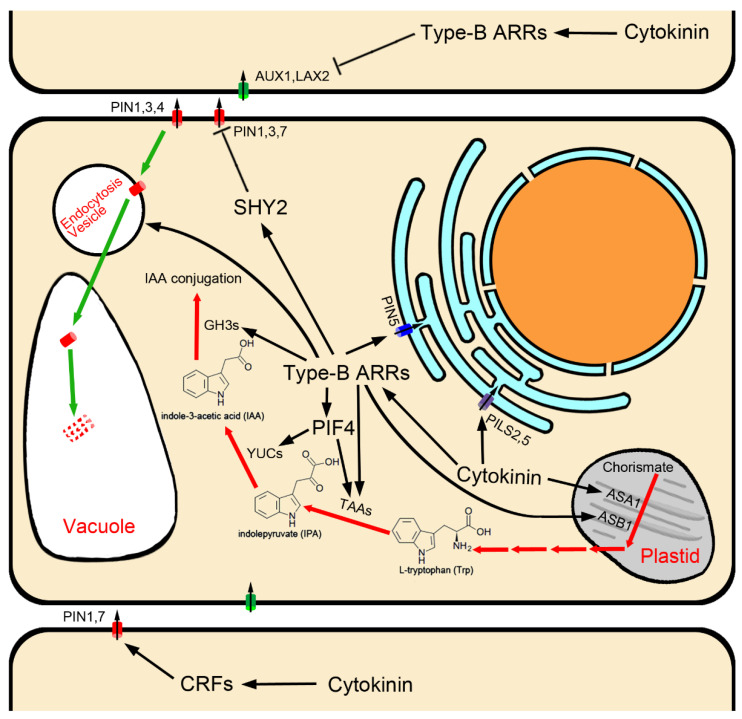
Auxin biosynthesis/metabolism and transport regulated by cytokinin.

**Figure 2 ijms-22-03874-f002:**
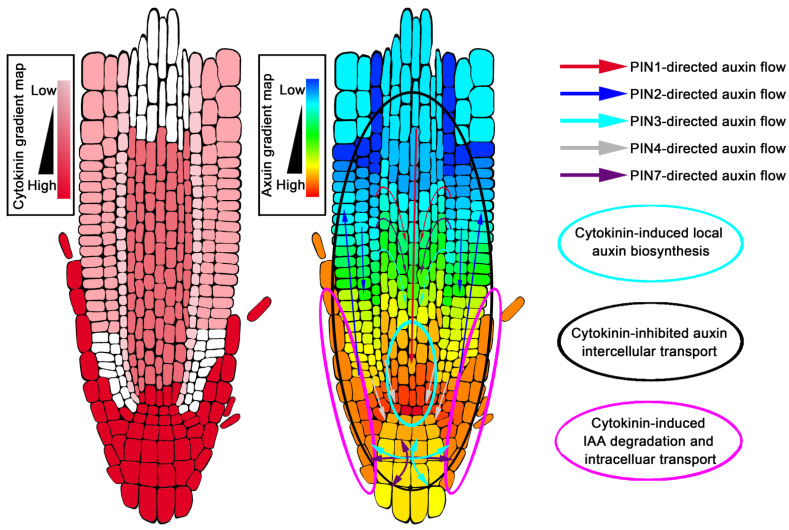
Cytokinin-controlled gradient distribution of auxin in *Arabidopsis* root tip.

## Data Availability

Not applicable.

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
