# Peer review of "Cytokinin-Controlled Gradient Distribution of Auxin in Arabidopsis Root Tip"

_ijms, 2021, doi:10.3390/ijms22083874_

Round 1

Reviewer 1 Report

The review is well written and organized. I do not feel that anything is missing or further needed. The only thing I would like to suggest is to remove the 'section' tags from the sub-headings.

Author Response

We gratefully appreciate for your valuable suggestion/ comment. We have deleted "section" tag in the revision.

Reviewer 2 Report

This is a well written and timely review of the cytokinin based regulatory system in the Arabidopsis root tip.    

I am a little surprised that this review contains just the one figure and think that additional figure(s) providing an overall view of the root tip and the flow of phytohormones around the quiescent centre could be added.  Perhaps the molecular cascades that constitute the regulatory mechanisms are also worthy of illustration to improve comprehension.  A reference to Fig 1 needs to be added into the text, at least in section 2.

Minor corrections:

Line 9-11 A key component regulating this growth and development is the finely tuned cross-talk between the auxin and cytokinin phytohormones.

Line 12  but also varies root growth in response

Line 13  light on the molecular

Line 16  mechanisms underlying the cytokinin-controlled auxin gradient in the primary root.

Line 22  usually located below

Line 39  Delete “by intense investigation”.

Lines 42, 56, 124,152, 244  The section headings are a bit confusing.  Eg 2. Cytokinin signal pathway, is all that is needed. Delete “Section  1:”

Line 43  . . in plants is similar to the bacterial . . .

Line 48  , which enters the nucleus

Line 57  genetic evidence

Line 80  Later, the authors found that . . .

Line 110  when the plant is treated . .

Line 125  In Arabidopsis, the GH3 . .

Line 127  IAA to IAA amino acid conjugates. The conjugation of amino acids with IAA is . . . 

Line 128  one that can be . .

Line 139  If the LRC is lost,  . . .

Line 150  . . amino acid conjugates, thus regulating the size of the root meristem.

Line 152/3  . .three protein families responsible for membrane trafficking:

Line 157  family proteins, involved in the

Line 159  directing

Line 169  in the cytokinin interaction

Line 172  depended

Line 179  polypeptide chain, the PINs family

Line 183  are expressed and active in the root tip.

Line 186  thereby modulating auxin distribution

Line 222  down-regulates PAT by

Line 232  LRC, where it plays a key role in maintaining

Line 235  genes act downstream

Line 239  CHIP assays showed

Line 245  In addition to the above mentioned  PINs

Line 248  There are seven known members of the PILS family.

Line 259  leading to reduced transcriptional regulation

Line 278  , however, the details require further study.

Line 282  which is derived from

Line 284  the meristem of the root tip reaches a stable size,

Line 294  auxin concentration is maximal at the QC of

Line 299  root tip depended on

Line 308  controls the flow of auxin to the root tip

Line 312  In addition, the auxin reflux mechanism gradually decreases auxin when

Line 317  (MS) assay was applied  to analyse

Line 318  each tissue of the root tip.  Cytokinin was found to be mainly concentrated

Line 319/20  cells, and in the epidermis of the root tip, forming a gradient down to a maxima at the root cap.

Line 236 indicated

Line 237  from the above-ground parts 

Line 333  In Section 2 we reviewed several

Author Response

We gratefully appreciate for your valuable suggestion/ comment. In the revision, we added Figure2 to provide an overall view of the root tip and the flow of phytohormones around the QC;  added reference to Figure 1 and Figure 2 into the text. In addition, the mistakes in MS have been corrected one by one.

This manuscript is a resubmission of an earlier submission. The following is a list of the peer review reports and author responses from that submission.